# CFD-Based Physical Failure Modeling of Direct-Drive Electro-Hydraulic Servo Valve Spool and Sleeve

**DOI:** 10.3390/s22197559

**Published:** 2022-10-06

**Authors:** Guoqin Huang, Juncheng Mi, Cheng Yang, Jin Yu

**Affiliations:** College of Mechanical and Vehicle Engineering, Chongqing University, Chongqing 400044, China

**Keywords:** direct-drive servo valve, CFD, physical failure model, erosion and wear, fluent simulation

## Abstract

Direct-drive electro-hydraulic servo valves are used extensively in aerospace, military and control applications, but little research has been conducted on their service life and physical failure wear. Based on computational fluid dynamics, the main failure forms of direct-drive electro-hydraulic servo valves are explored using their continuous phase flow and discrete phase motion characteristics, and then combined with the theory of erosion for calculation. A mathematical model of the direct-drive electro-hydraulic servo valve is established by using Solidworks software, and then imported into Fluent simulation software to establish its physical failure model and carry out simulation. Finally, the physical failure form of the direct drive electro-hydraulic servo valve is verified by the simulation results, and the performance degradation law is summarized. The results show that temperature, differential pressure, solid particle diameter and concentration, and opening degree all have an impact on the erosion and wear of direct-drive electro-hydraulic servo valves, in which differential pressure and solid particle diameter have a relatively large impact, and the servo valve must avoid working in the range of high differential pressure and solid particle diameter of 20–40 um as far as possible. This also provides further theoretical support and experimental guidance for the industrial application and life prediction of electro-hydraulic servo valves.

## 1. Introduction

Direct-drive electro-hydraulic servo valves are widely used because of their low leakage, good control characteristics, high actuation force, low energy consumption and high safety. The output flow or pressure of an electro-hydraulic servo valve is controlled by an electrical input signal and is mainly used in high-speed closed-loop hydraulic systems to realize the control of position, speed and force. The electro-hydraulic servo valve is one of the core servo control elements in a closed-loop control system, which can convert small electrical signals into high-power hydraulic signals [1].

The direct-drive electro-hydraulic servo valve is primarily driven by a solenoid-type linear force motor and uses an integrated circuit to provide feedback on the position of the control spool. The alignment spring keeps the spool in the neutral position, while the linear force motor allows the spool to move in both directions by overcoming the spring force and balancing in a new position [2]. When the direct-drive servo valve is working, an electrical signal is added to the integrated controller and the integrated circuit; then it forms a pulse-width modulated current in the linear motor and the force motor drives the spool to a certain displacement, while the oscillator excites the spool position sensor. The deviation signal obtained from the comparison changes the input current until the spool position reaches the set position.

With the wide application of direct-drive electro-hydraulic servo valves in many fields, their safety and reliability are paid more and more attention by the industry. The spool sleeve, as a key component of the direct-drive electro-hydraulic servo valve, is most sensitive to contamination related to the fluid. The working condition of the direct-drive electro-hydraulic servo valve also affects the safety and reliability of the entire hydraulic control system, so it is necessary to analyze and discuss the impact on its working performance and service life. Zhang Kun et al. [3] of Beijing University of Aeronautics and Astronautics, China, used the sliding valve assembly of an electro-hydraulic flow servo valve as the object and developed a mathematical model of fluid erosion of the spool valve sleeve based on fluid dynamics and Edwards’ erosion and wear equation for Fluent simulation. Zhang Hong et al. [4] of Taiyuan University of Technology in China studied a high water-based planar pilot valve with a conventional positive top rod structure, using the low Re number k-ε turbulence model proposed by Jones, Launder and Edwards et al. and sand impact on metal surface experiments in order to establish a simulation model. In addition, Amirante [5] developed a new structural design for the spool to reduce wear by reducing the hydrodynamic forces. Barman P. [6] developed a new model of a slide valve and used the STAR-3D simulation software to simulate it and obtain pressure clouds of the flow field inside the valve body in the simulation results. The Italian scholars Borghi M et al. [7] used the integral method to calculate the pressure values on the spool surface at different opening degrees, thus inferring the proportion of hydrodynamic forces under transient and steady-state conditions.

Thus, it can be seen that extensive research has also been carried out on the degradation of various types of hydraulic valves around the world. However, little research has been carried out on the wear of spool bushings for direct-drive servo valves. The usual forms of failure of servo valve spool bushings are erosion failure, sludge failure, jamming failure and corrosion failure, among which the main form of failure is erosion failure. Regarding the study of erosion failure, Finnie I. [8] conducted theoretical and experimental studies on the erosion and wear mechanism of plastic materials and proposed a micro-cutting theory of erosion and wear of plastic materials, from which an empirical formula for its erosion was derived, and the micro-cutting theory at low incidence angles was verified through experiments. However, the disadvantage of this theory is that when particles are eroded at large angles, there is a large error between the equation of wear and the experimental results. Bitter [9] creatively proposed a deformation wear theory based on microcutting wear and deformation wear mechanisms, and this theory completes the microcutting theory. According to this theory, that material wear is the result of a combination of micro-cutting and deformation wear, with cutting wear dominating when solid particles hit the surface of an object at a small angle, and deformation wear dominating when particles hit the surface of an object at a large angle. Tilly [10] found in the experiment by high-speed camera brittle particles and plastic material collision wear that some cracked particles and then the material will cause a second erosion wear, and for the first time put forward a secondary erosion theory that the total amount of erosion wear is equal to the sum of the two stages of erosion. Levy [11], through a large number of experiments and data analysis, concluded that the particle erosion was caused by the loss of material mass; first, the particle impacts on the surface of the material to form a thin sheet, followed by the thin sheet in the particle under the repeated extrusion of plastic deformation, and finally into a fine sheet away from the surface of the material, and thus the extrusion forging theory for plastic material erosion wear was put forward.

For further research on the theory of erosion wear, in recent years, Solecka et al. [12] carried out erosion wear experiments using Al2O3 as solid particles on the surface of the alloy after the cold metal transfer (CMT) process and investigated the specific mechanism of such erosion wear at two kinds of temperatures. However, this paper only discusses the principle and process of erosion wear, and there is no specific discussion of the changes in erosion and wear brought about by the many different temperatures. While Liu Y et al. [13] investigated the effect of abrasive hardness on erosion wear using abrasive air jets, Zambrano [14] conducted solid particle erosion wear studies using different abrasives and materials of different hardness and defined an abrasive/material hardness ratio to distinguish between light, medium, and moderate definitions of erosion wear. Bhosale et al. [15] investigated the resistance to erosion and wear at high temperatures of 800, 650 and 500 °C for coating materials subjected to atmospheric plasma spraying and high speed oxyfuel processes. Nayak et al. [16] studied the wear surface morphology of waste marble dust to investigate the mechanism of its erosion loss and calculated its erosion wear efficiency. Xu L J et al. [17] verified the use of nitrogen and chromium as well as multi-scale carbide incorporation into parts in industrial applications, thus providing effective resistance to erosion wear. Ma Y P et al. [18] conducted erosion wear tests on magnesium alloys before and after surface diffusion alloying treatment and demonstrated the effective resistance to erosion wear after surface diffusion alloying treatment. Kumar et al. [19] studied the materials and processes of 7YSZ coatings regarding erosion wear at room temperature and at high temperatures of 850 °C. Most of these studies focused on the influence of material aspects of the target object on erosion wear, but the influence of other factors on erosion wear is rarely considered and comparatively analyzed. In addition, Boggarapu et al. [20] studied the effect of new polymer matrix composites (PMCs) subjected to erosion and wear, describing the effect of three factors, namely the impact velocity, erosion characteristics and impact angle of solid particles, on the erosion and wear rate of this material and its failure mechanism. However, the scope of application is only roughly defined and focuses on combining the different variation factors under this material, without systematically studying and analyzing the specific variation patterns of erosion and wear under different factors. Furthermore, Ou G F et al. [21] performed a numerical prediction of the erosion wear of the pipe and a validated model of the particle trajectory and erosion rate was calculated. Zhang W G et al. [22] studied the erosion wear condition of rotors made of carbide materials from different scouring angles and abrasive sizes. Demirci et al. [23] designed a new alternative test stand for erosion wear at high temperatures, but this was only designed for high temperatures at 700 °C, while it was not discussed for medium to high temperatures or even lower.

It can be seen from the above literature that in the current industrial field, there is little research on the spool sleeve wear of direct-drive electro-hydraulic servo valves and the factors influencing the main form of failure, erosion and wear, and the combination of the two; especially the effect of parameters such as differential pressure, flow rate and opening on the erosion and wear of the spool sleeve of direct-drive servo valves is still unclear. Therefore, it is necessary to further study the erosion and wear of spools for direct-drive servo valves. The purpose is to discuss systematically and specifically the mechanism of erosion and wear of direct-driven electro-hydraulic servo valve spools and sleeves under different factors, and the influence of each factor on erosion and wear. In this paper, based on the knowledge of fluid mechanics, the fluid control equations inside the valve are established. Based on the existing erosion and wear model, a mathematical model of the erosion and wear of the spool sleeve for direct-drive servo valves is established, and then the fluid domain inside the servo valve is established using Solidworks, and then simulated with Fluent. The study of the erosion wear efficiency of the servo valve spool sleeve under different structural and flow parameters such as different opening degrees, contaminated particle diameters and differential inlet and outlet pressures is carried out based on simulation results. This article focuses on the theoretical and simulation studies of the erosion and wear of servo valve spools and sleeves. Figure 1 shows the overall work flow chart.

## 2. Physical Model for Servo Valve Wear Faults

### 2.1. Basic Introduction to Servo Valves

In terms of hydraulic characteristics, the flow in a direct-drive servo valve is generally completely turbulent. The flow of oil containing solid contaminated particles inside the valve is a liquid-solid two-phase flow, in which the oil is the continuous phase and the contaminated particles are the discrete phase. In terms of construction, most valves are made up of two main components, the spool and the body, as well as other components that control the movement of the spool. In principle, basically all valves achieve their respective roles in the hydraulic system by changing the size of the valve orifice. This paper focuses on the flow of fluid inside a direct-drive servo valve supplied by the Chinese Institute of Aeronautical General Technology, the operating principle of which is shown below for the spool valve sleeve.

As shown in Figure 2, when the force motor has no command input, the spool is in the hydraulic zero position, the throttle window is not opened and the control cavities A and B are not in communication with the oil feed and return oil.

As shown in Figure 3, the force motor inputs a positive command pulse width modulation current, generates positive driving force, and drives the spool to a certain displacement and pushes the spool to the left, while the oscillator makes the spool position sensor excited; after demodulation of the spool’s actual position signal and set position signal for comparison, the deviation signal obtained by comparison changes the input current’s size, until the spool position reaches the set position. At this time, the left window of the oil inlet P is opened as the oil inlet, so that the control cavity A communicates with the inlet cavity; the right window is opened as the return port, so that the control cavity B communicates with the return cavity.

Similarly, as shown in Figure 4, the force motor inputs negative command pulse width modulation current, produces a negative driving force, drives the valve spool to occur at a certain displacement, and pushes the valve spool to the right movement at this time, so that the right window of the oil inlet P opens as the oil inlet, and the control cavity B communicates with the inlet cavity; the left window opens as the return port, so that the control cavity A communicates with the return cavity.

### 2.2. Spool Valve Sleeve Erosion Wear Model

In the flow of liquid-solid fluids, the erosion of wall material by solid particles is a common form of wear, defined as the loss of surface material when a solid surface comes into contact with a fluid containing solid particles in relative motion, usually less than 1 mm in diameter and with an erosion velocity of up to 550 m/s. The trajectory of contaminated particles in the flow field influences the flow field and the prediction of wear on the boundary surface. In the working process, because the electro-hydraulic servo valve working medium has some solid pollution particles, the movement of solid pollution particles in the flow field is affected by a variety of stresses such as compressive stress and shear stress, which are then combined with Newton’s second law to get:(1)$$m\frac{dv}{dt}=F_{D}+F_{g}+F_{X}$$
where *F_D_*—travelling forces on the particles;

*F_g_*—gravity forces;

*F_X_*—additional forces, e.g., Brownian force, Saffman lift, etc.

In the electro-hydraulic servo valve’s work, solid pollution particles follow the medium together, and constantly hit the spool or sleeve; the solid pollution particles’ momentum and energy are reduced at the same time, but also cause the wear of the spool sleeve. The state change of particles is often determined by the bounce coefficient, which is a function of the impact angle of the particles, i.e., the ratio of the normal velocity component before and after the particles hit the wall and the tangential velocity component before and after the particles hit the wall, respectively. In this paper, the Forder formula is used to calculate the normal and tangential bounce coefficients derived from experiments on quartz sand particles eroding a carbon steel wall [24]:(2)$$\in_{N}=0.993-0.0307\alpha_{p}+4.75\times 10^{-4}\alpha_{p}^{2}-2.61\times 10^{-6}\alpha_{p}^{3}$$
(3)$$\in_{T}=0.998-0.029\alpha_{p}+6.43\times 10^{-4}\alpha_{p}^{2}-3.56\times 10^{-6}\alpha_{p}^{3}$$
where α*_p_*—the angle of incidence of the particle;

*∈_N_*—ratio of the velocity component normal to tangential before the particle strikes the wall of the spool sleeve.

*∈_T_*—the ratio of the normal to tangential velocity components of the particles after they hit the wall of the spool sleeve.

The erosion wear rate of the spool sleeve was numerically simulated using the erosion wear model proposed by Edwards based on grit erosion experiments on carbon steel and aluminium surfaces. Equation [25] is as follows:(4)$$R_{\mathrm{erosion}}=\sum_{p=1}^{N_{p}}\frac{m_{p}C\left(d_{p}\right)f\left(\alpha \right)v^{b\left(v\right)}}{A_{face}}$$
where *p*—the number of solid contaminating particles;

*A_face_*—the unit area of solid contamination particles impacting the wall of the spool valve sleeve, m^2^;

*α*—the impact angle between the trajectory of the solid pollution particles and the wall of the spool valve sleeve, °;

*b(v)*—the function of the relative motion of the solid pollution particles, generally taken as 0.2 to 0.4;

R_erosion_—the erosion and wear rate of the spool sleeve, kg/(m^2^·s);

*C(d_p_)*—as a function of the diameter of the solid contaminated particles, also representing a flow coefficient, i.e., the flow area of the valve orifice to the area of the valve orifice;

*v*—velocity of relative motion of solid pollution particles, m/s;

*m_p_*—mass flow rate of solid contaminated particles, kg/s;

*f(**α)*—impact angle function of the solid pollution particles.

Metal materials are usually plastic materials; When solid particles hit the surface of a material, they will cause material loss and attrition; if the nature of the surface material is different, the form and pattern of wear will be different on a microscopic level, while plastic materials will be deformed by the impact of solid particles but their integrity will not be destroyed; generally, the maximum damage occurs at an angle impact of 20° to 30°;

*d_p_*—the diameter of the solid pollution particles, m.

The relative changes in impact angle function and impact angle for solid contaminated particles [26] are shown in Table 1.

Equation (4) defines the physical significance of the wear quality of the spool sleeve in relation to the erosion rate in erosion wear. By applying Equation (4) to the fluid dynamics software Fluent, the erosion wear rate can be simulated to determine the performance degradation and life prediction of the spool sleeve. If the maximum allowable wear distance of the prongs is *l*_max_, then the theoretical mean life can be simulated from the maximum erosion wear rate, as shown in Equation (5):(5)$$T=\frac{l_{\max}}{R_{\max}/\rho_{s}}$$

### 2.3. Theory Related to Computational Fluid Dynamics

Computational Fluid Dynamics (CFD) is a method of simulating numerical calculations and graphic displays using a computer, which allows for the visual and quantitative representation of fluid motion patterns in time and space.

There are many methods to simulate numerical calculations in CFD software, mainly including the Finite Difference Method, the Finite Element Method and the Finite Volume Method.

The flow of fluid also follows the three conservation laws of mass, momentum and energy. If the different components of a fluid react and interact with each other, the law of material conservation is also observed, and if the fluid is in a turbulent state, the turbulence model theory is followed. On the basis of the three conservation laws, CFD has established three kinds of mechanical equations of fluid: the continuity equation, the differential equation of motion and the energy conservation equation.

#### 2.3.1. Continuity Equation

From the perspective of flow field, according to mass conservation, the total mass of a fluid flowing through a certain space remains constant over dt [27]. According to this law, the corresponding continuity equation can be derived as:

Transient compressible fluids:(6)$$\frac{\partial \rho }{\partial t}+\frac{\partial \left(\rho u\right)}{\partial x}+\frac{\partial \left(\rho v\right)}{\partial y}+\frac{\partial \left(\rho w\right)}{\partial z}=0$$

Incompressible fluids:(7)$$\frac{\partial \left(u\right)}{\partial x}+\frac{\partial \left(v\right)}{\partial y}+\frac{\partial \left(w\right)}{\partial z}=0$$

Steady state fluids:(8)$$\frac{\partial \left(\rho u\right)}{\partial x}+\frac{\partial \left(\rho v\right)}{\partial y}+\frac{\partial \left(\rho w\right)}{\partial z}=0$$

Vector form:(9)$$\frac{\partial \rho }{\partial t}+div\left(\rho \vec{u}\right)=0$$
where *ρ*—density;

*t*—time;

$\vec{u}$—velocity vector;

*u*—the component of the velocity vector in the *x*-direction;

*v*—the component of the velocity vector in the *y*-direction;

*w*—the component of the velocity vector in the *z*-direction.

#### 2.3.2. Differential Equations of Motion

The relationship between the fluid in motion and the external forces on it follows the law of conservation of momentum, i.e., the sum of all the external forces acting on a given tiny fluid system at a given time is equal to the product of the total mass of the system and its acceleration. The differential equation of motion is the mathematical expression of this law.
(10)$$\frac{\partial \left(\rho u\right)}{\partial t}+div\left(\rho u\vec{u}\right)=-\frac{\partial p_{0}}{\partial x}+\frac{\partial \tau_{xx}}{\partial x}+\frac{\partial \tau_{yx}}{\partial y}+\frac{\partial \tau_{zx}}{\partial z}+F_{x}$$
(11)$$\frac{\partial \left(\rho v\right)}{\partial t}+div\left(\rho v\vec{u}\right)=-\frac{\partial p_{0}}{\partial y}+\frac{\partial \tau_{xy}}{\partial x}+\frac{\partial \tau_{yy}}{\partial y}+\frac{\partial \tau_{zy}}{\partial z}+F_{y}$$
(12)$$\frac{\partial \left(\rho w\right)}{\partial t}+div\left(\rho w\vec{u}\right)=-\frac{\partial p_{0}}{\partial z}+\frac{\partial \tau_{xz}}{\partial x}+\frac{\partial \tau_{yz}}{\partial y}+\frac{\partial \tau_{zz}}{\partial z}+F_{z}$$
where *p*_0_ is the static pressure on the fluid; $\tau_{xx}$, $\tau_{yx}$ and $\tau_{zx}$, etc. are the components of the viscous shear stress τ on the surface of the fluid microcluster, and *F_x_*, *F_y_* and *F_z_* are the mass forces on the fluid microcluster; if the only mass force is gravity and the z-axis is vertical upwards, then we have *F_x_* = 0, *F_y_* = 0 and *F_z_* = −*ρ**g*.

#### 2.3.3. Conservation of Energy Equation

The fluid will also be affected by temperature changes in the process of movement, so it must also satisfy the law of conservation of energy, i.e., the increase and decrease of energy in the fluid microcluster is equal to the net heat flow into the microcluster and the sum of how much work the mass force and pressure does on the microcluster.

The energy *E* of a fluid usually includes three types: internal energy, kinetic energy and potential energy. However, because of the complexity of fluids, the change in kinetic energy is neglected to obtain the equation of energy conservation with respect to internal energy *i*. The internal energy *i* and the temperature *T* satisfy *i* = *C_p_T*. Using the temperature *T* as a variable, the following equation for the conservation of energy of internal energy is obtained:(13)$$\begin{array}{l} \frac{\partial \left(\rho T\right)}{\partial t}+\frac{\partial \left(\rho uT\right)}{\partial x}+\frac{\partial \left(\rho vT\right)}{\partial y}+\frac{\partial \left(\rho wT\right)}{\partial z}\\ =\frac{\partial }{\partial x}\left(\frac{k}{c_{p}}\frac{\partial T}{\partial x}\right)+\frac{\partial }{\partial y}\left(\frac{k}{c_{p}}\frac{\partial T}{\partial y}\right)+\frac{\partial }{\partial z}\left(\frac{k}{c_{p}}\frac{\partial T}{\partial z}\right)+S_{T} \end{array}$$
where: *T*—temperature;

*S_T_*—the internal heat source of the fluid and the fraction of mechanical energy converted to heat, i.e., the viscous dissipation term;

*C_p_*—the specific heat capacity;

*k*—the heat transfer coefficient of the fluid.

## 3. Spool Valve Sleeve Erosion Wear Simulation Analysis

According to the given spool sleeve model and its main material 9Cr18, as well as the relevant literature, combined with the factors affecting particle erosion wear and the theoretical description in Section 2, the five factors of opening degree, temperature, oil solid particle diameter, oil solid particle concentration, and servo valve inlet and outlet differential pressure are selected, all of which affect the fluid flow rate, flow velocity, momentum and energy, and thus cause the change in spool sleeve erosion wear. For the opening degree, temperature, oil solid particle diameter, oil solid particle concentration, servo valve inlet and outlet pressure difference and other five factors are considered for the spool valve sleeve erosion wear simulation; in addition, because of the oil contamination level, the trend of change is somewhat different for mixed particles and a single particle, so the simulation results out of the error will have a large error, and it is not considered. In this section, the three-dimensional model of the spool sleeve of the direct-drive electro-hydraulic servo valve is established, and the continuous phase model and discrete phase model of the Fluent software are used to simulate the erosion and wear of the direct-drive electro-hydraulic servo valve so as to obtain the erosion and wear results of the spool sleeve for different opening degrees, different contamination particle concentrations and different contamination particle diameters.

### 3.1. Model Building and Fluid Domain Extraction

Solidworks software is used to model the spool sleeve of the direct-drive electro-hydraulic servo valve. Based on the dimensions of the direct-drive electro-hydraulic servo valve provided by the Chinese Institute of Aeronautical Technology, this model was set at 21.87 mm in diameter and 96 mm in length. Then, the simulation model in Solidworks was imported through the SpaceClaim component in ANSYS Workbench, and the flow field model formed by the spool sleeve was extracted and meshed in the meshing component as shown in Figure 5.

As the spool sleeve inlet is throttled, the main wear area is near the inlet, so when meshing, the mesh of this part is set to a smaller size to facilitate the obtaining of more realistic data; the minimum mesh size is 0.1 mm and the minimum mesh quality is 0.3. The entire computational domain can represent the complete 3D geometry of the sleeve, with only one plane of symmetry. In order to facilitate the numerical simulation of the direct-drive spool valve sleeve, the following assumptions were made in the simulation:(1)Since the diameter of the contaminated particle is in the micron range, it was simplified to a circular sphere, and the effect of the particle’s own rotation on the particle trajectory and the effect of erosion were ignored;(2)The roughness of the spool sleeve surface was assumed to be zero;(3)When performing simulations with non-temperature variation factors, the effect of changes in hydraulic oil temperature on the flow in the valve is not considered, i.e., a constant oil temperature of 300 K is assumed.(4)The oil is incompressible and has no heat exchange;(5)As the temperature rises between 20 and 120 °C at room temperature, the effect on gas heat capacity is greater, but the effect on liquid heat capacity is less, so this simulation does not consider the effect of heat capacity of the working oil as the temperature rises for the time being.

Main simulation parameter settings:(1)Fluid material: China No. 15 aviation oil, density 855 kg/m^3^; dynamic viscosity;(2)Discrete phase parameters: Particle size or diameter; particle rate is usually set to 10 m/s;(3)Viscosity model: standard k-ε viscosity model; and discrete phase model;(4)Boundary conditions: Pressure inlet boundary condition set to 28 MPa and the outlet boundary condition to 14 Mpa when the pressure is constant; the wall is set by the DPM bounce model; turbulence intensity is set at 5% and the hydrodynamic diameter at 8 mm; the temperature at set at 300 K when performing simulations with influences other than temperature;(5)Number of iterations: set to 500 steps.

### 3.2. Spool Sleeve Wear Analysis for Different Opening Degrees

Measurements by Solidworks software were known: the maximum opening of the spool sleeve was 0.5 mm, so the Solidworks software was used to derive five models with spool sleeve openings of 0.1, 0.2, 0.3, 0.4 and 0.5 mm at a solid particle diameter of 5 um. The simulation was carried out using Fluent for each of the five models.

#### 3.2.1. Valve Port Flow Velocity Analysis

According to the simulation parameters and assumption settings, the liquid flow velocity of the valve port at five different openings is derived by the simulation accordingly as shown in Figure 6:

As can be seen from the above five velocity clouds, the maximum velocity of fluid flow near the valve port exceeds 200 m/s. Moreover, the velocity is lower at some right angles, for example, at the right angle of the small hole of the inlet port at small openings and at the right angle area where the large diameter of the spool is connected to the small diameter. In the case of the small opening, the fluid flow through the valve orifice is also small, with a positive correlation between the range of velocity changes and the flow rate, and the maximum velocity shot out of the fluid also appears to be less due to the too-small overflow area of the valve orifice. The maximum fluid velocity occurs near the valve opening, where the velocity changes are also more violent. When the flow of fluid through the valve port decreases gradually, the flow will increase when it reaches the valve core diameter. Generally speaking, when the differential pressure between the inlet and outlet is certain, with the increase in the opening, the fluid flow at the valve port is increasing, so the flow rate of the oil at the small area of the valve sleeve inlet is also increasing. When the valve opening increases, the irregularity of the particle trajectory from the inlet to the valve side increases and the number of collisions with the wall increases, so it is clear that the valve opening affects the erosion wear rate of the spool sleeve.

#### 3.2.2. Effect of Opening on Erosion Wear

The results of the simulation, which yielded the spool sleeve erosion rate at five different openings, are shown in Figure 7:

From the above (a) to (e), five clouds can be seen; erosion wear occurs in the liquid flow from the valve mouth after the impact to the spool small diameter, the spool valve sleeve throttle edge and nearby. From (f), it can be seen that the maximum erosion rate increases with the opening degree and then decreases, and reaches a maximum value of 7.29 × 10^−5^ when the opening degree is 0.3 mm; the minimum value is 1.05 × 10^−5^ when the opening degree is 0.1 mm. The maximum erosion rate increases more before the opening degree of 0.3 mm and decreases more slowly afterwards, and the difference between the maximum erosion rate when the opening degree is 0.3 mm and 0.4 mm is smaller. Combined with the fluid flow analysis under the opening simulation in Section 3.2.1 above, the velocity distribution of the fluid in the valve and the trajectory of the solid contamination particles are different for different orifice openings, which will cause differences in the location of the spool sleeve erosion wear and the erosion wear rate. However, on average, the maximum erosion rate tends to increase with increasing openness. 

### 3.3. The Effect of Oil Temperature on Erosion Wear

As the change in fluid temperature will cause the change in fluid viscosity, the dynamic viscosity of Chinese No. 15 aviation hydraulic fluid at different oil temperatures was obtained by reviewing the literature and then converting it as shown in Table 2:

The change in temperature leads to a change in fluid viscosity and will convert the endothermic and mechanical energy of the fluid into a thermal energy component, combined with the thesis Equations (2)–(13); then, according to the law of conservation of energy, the internal and mechanical energy of the fluid and solid particles change, which leads to changes in erosion wear. In order to obtain more accurate simulation results, the oil temperature is divided into 6 groups of 20, 40, 60, 80, 100 and 120 °C, starting with the room temperature setting of 20 °C as a reference in the table above. In Fluent, the same 0.5 mm simulation model was used, with the diameter and concentration of the particles kept constant and the inlet and outlet pressures kept constant. The energy equation was opened in Fluent and the kinetic viscosity values from Table 2 were entered into the fluid setup. At the same time, the outlet and inlet temperatures were set to the above eight different sets of temperatures for the simulation. The simulation results are shown in Figure 8 below:

According to Figure 8a–f, it can be seen that at room temperatures, the erosion mainly occurs at the oil inlet, i.e., at the change in area of the oil inlet hole of the sleeve; with the increase of temperature, the oil inlet on the sleeve is still eroded, but the maximum erosion rate has been transferred to the small diameter shoulder of the spool, and there is also a certain amount of erosion at the side of the oil inlet. It can also be seen from Figure 8g that the overall maximum erosion rate of the spool sleeve tends to increase as the temperature rises, where the change in the maximum erosion rate is a linear slow increase when the oil temperature is below 60 °C, and a rapid increase when it is above 60 °C. Under standard operating conditions, the oil temperature is 40 ± 8 °C; from the simulation results, in this range, the corrosion rate changes little, and the high oil temperature also has certain damage to other components in the experiment.

### 3.4. The Effect of Contaminated Particles on Erosion Wear

#### 3.4.1. Effect of Contaminated Particle Concentration on Erosion Wear

Keeping the particle diameter constant, the particle concentration was increased, i.e., the pollutant concentration was increased. The model with a 0.5 mm opening and a solid particle diameter of 5 um was chosen for the simulation and the particle concentration was increased by a factor of 2, 4, 8 and 16. In the Fluent parameter settings, the particle flow rate could be changed by increasing the particle flow rate by a factor of 2, 4, 8, 16 and 32, resulting in the simulation results shown in Figure 9.

It can be seen from Figure 9 that, when the particle diameter is constant, the maximum erosion rate increases as the particle concentration increases exponentially, and the maximum erosion rate increases by a factor close to that of the particle concentration. This is because, when the particle diameter is constant, increasing the particle concentration is equivalent to having more particles impacting on the spool sleeve in the same amount of time, so that the maximum erosion rate and the particle concentration are a primary function of the rate of change, i.e., when the particle concentration increases by a multiple, the maximum erosion rate also increases by the same multiple.

#### 3.4.2. Effect of Contaminated Particle Diameter on Erosion Wear

According to Chinese GJB 420B-2015 “Classification of solid contamination in aviation working fluids” [28], the classification is based on the size, particle number and distribution of solid contaminants in the fluid, and the content of solid contaminants per unit volume of working fluid. The standard will be a specific range of particle diameters are denoted by the letters A, B, C, D, E and F. Simulations are carried out by increasing the particle diameter to obtain results on the effect of particle diameter on the erosion rate. The particle diameters were taken to be 5 um, 10 um, 20 um, 35 um, 55 um and 100 um under six diameter ranges, and the same model with a 0.5 mm opening was used for the simulation analysis. In the simulation parameters, the particle flow rate can be changed. The results are shown in Figure 10.

It can be seen from Figure 10 that when maintaining the pollution level and increasing the particle diameter, the maximum erosion rate of the spool valve sleeve is a sharp increase followed by a slow decrease, and reaches a maximum when the solid particle diameter is 20 um. This is due to the fact that after the particle diameter increases to a certain level, the gravity of the individual particles will have a greater impact on their movement; when the particle diameter is small, the movement of the particles is mainly influenced by the hydrodynamic force generated by the oil.

### 3.5. Effect of Differential Inlet and Outlet Pressures on Erosion and Wear

It can be seen from the previous section that, since the rated supply pressure of the servo valve is 28 ± 0.5 MPa and the outlet pressure is set to 14 MPa, in order to derive the variation of the erosion rate of the spool sleeve at different operating pressures, the differential inlet and outlet pressures were taken to be 2 MPa, 4 MPa, 6 MPa, 8 MPa, 10 MPa, 12 MPa, 14 MPa and another seven groups. The same simulation model with an opening of 0.5 mm and a particle diameter of 5 um was chosen, ignoring the effect of temperature. In Fluent, it was only necessary to change the pressure at the oil port; the outlet pressure remained unchanged and the rest of the parameters and conditions already set remained unchanged. The simulation results are shown in Figure 11a–g below; in the same way for the rest of the openings such as 0.1 mm and 0.3 mm, the simulation results are shown in Figure 11h below:

As can be seen from Figure 11a–g, other conditions being the same, the distribution of erosion is similar for different inlet and outlet differential pressures, mainly on the sharp side of the sleeve at the valve port and on the sharp side of the spool control, but the location of the severe erosion points is different. With the increase in pressure difference, the area with severe erosion wear gradually changes from the simple sleeve inlet to the sleeve inlet and the shoulder of the valve element with small diameter. In addition, there is a certain amount of erosion wear in the axial section of the large diameter shoulder of the spool and near the wall of the sleeve near the oil inlet. The reason the serious erosion point will be shifted is that as the pressure difference between the inlet and outlet increases, the inlet fluid velocity increases, and under the action of dragging force and inertia, most of the particles move to the upper part, thus making the upper part of the particles more likely to collide with the wall. At the same time the erosion wear also increases with the increase in differential pressure. According to Figure 11h, the maximum erosion rate of the spool sleeve tends to increase as the differential pressure increases, and the trend is slower until the differential pressure is 8 MPa, then more rapid. There are some deviations in the individual data, which should be due to the calculation error caused by the simulation, and the overall approximation is a certain quadratic function.

## 4. Discussion

According to CFD, the erosion analysis of the spool sleeve model under different variables was established. Comparing the above five sets of simulation data, due to the rectangular inlet of the fluid flow, there is a certain amount of sleeve erosion near these corners; for the spool, the erosion wear rate of the sharp edge of the control surface of the slide valve sink groove is significantly greater than that of the sharp edge of the control surface of the convex shoulder. The erosion rate is influenced by the contaminated solid particles of the oil, with the increase in the contaminated solid particles parameter, the erosion rate changes more obviously, and for the particle sizes of 20 um and 35 um, the erosion wear is the most serious. The erosion rate is less affected by the change in opening degree, and increases and then decreases with the increase in opening degree. For fluid temperature, the erosion rate is approximately linear after 20°C, but compared with other influencing factors, temperature has less influence on erosion and wear; considering the difficulty of temperature control during the experiment and the sensitivity of the remaining hydraulic components to temperature and other factors, it can be ignored. As for the differential pressure, when the differential pressure increases, the increasing trend in the erosion rate is an approximately quadratic function.

## 5. Conclusions

Based on the multi-factor enumeration method and theoretical analysis, the physical model of spool sleeve failure is established, and it is considered that erosion wear is the main form of spool sleeve failure. Considering that the erosion wear failure form of the spool valve sleeve is closely related to the boundary conditions and various working parameters, based on the classical finite difference method and the CFD theory related to flow dynamics and taking the spool valve sleeve of the direct-drive flow servo valve, which is commonly used and relatively simple in the aerospace field, as an example, Fluent was used to simulate the flow field dynamics of the direct-drive servo valve core and sleeve under the condition of multiple parameters. The simulation analysis mainly considers the four most influential operating parameters: opening degree (corresponding to the flow rate of the valve), medium temperature, oil contamination with solid particles and operating pressure. In this paper, the internal motion characteristics of the direct-drive servo valve and the erosion wear of the slide valve sleeve are studied, discussed and verified through simulation. The results and laws obtained of the simulation study give the general influence trend of different factors on the erosion wear of the spool valve sleeve. It can be used to establish a mathematical prediction model for more accurate quantitative research in the future, and can guide the structural optimization and life prediction of the spool valve sleeve of the direct-drive servo valve. If the mathematical model of wear and life of direct-drive servo valves can be further improved, the physical failure discussion and performance degradation prediction of direct-drive servo valves can be further enhanced.

## Figures and Tables

**Figure 1 sensors-22-07559-f001:**
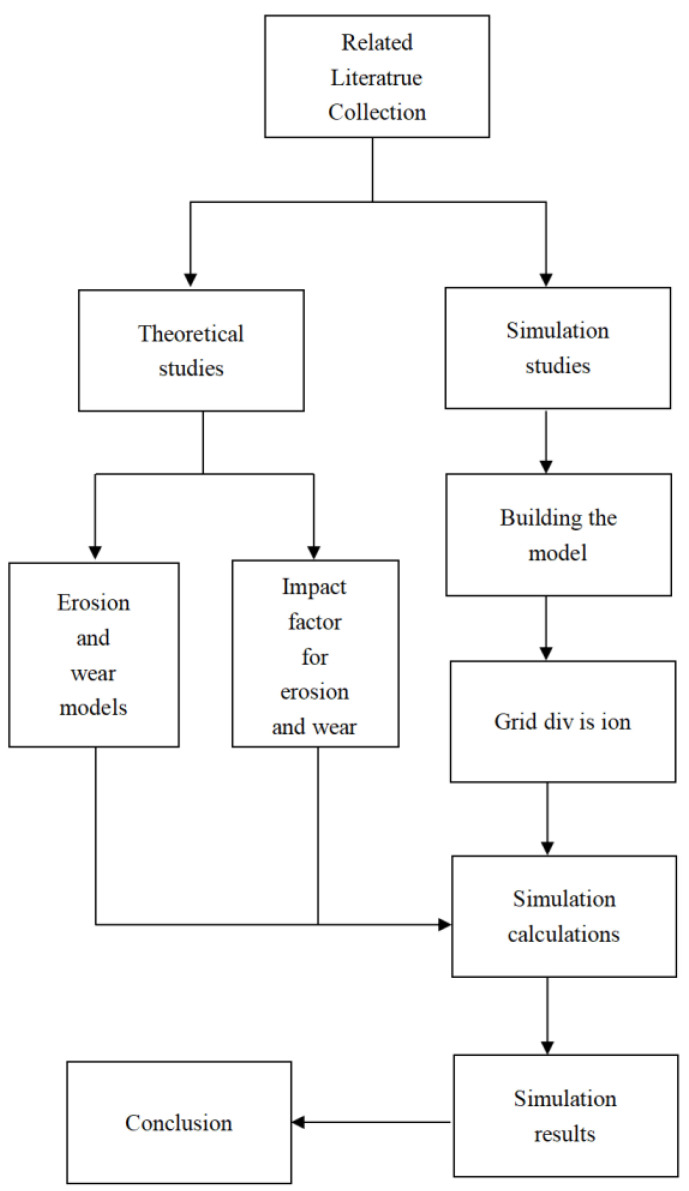
Overall work flow chart.

**Figure 2 sensors-22-07559-f002:**
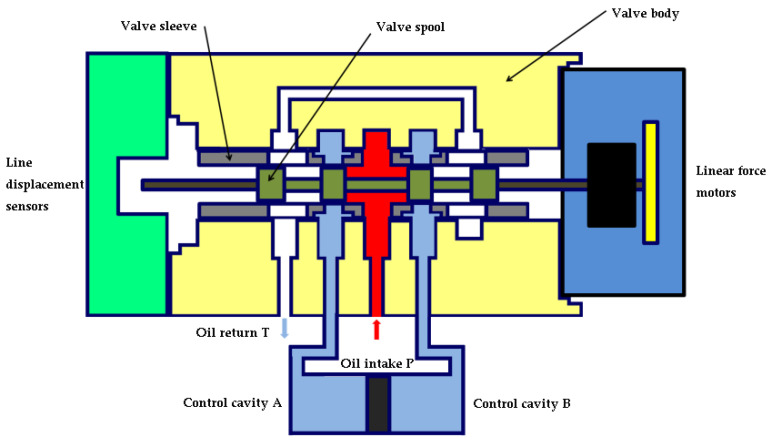
Servo valve zero position state.

**Figure 3 sensors-22-07559-f003:**
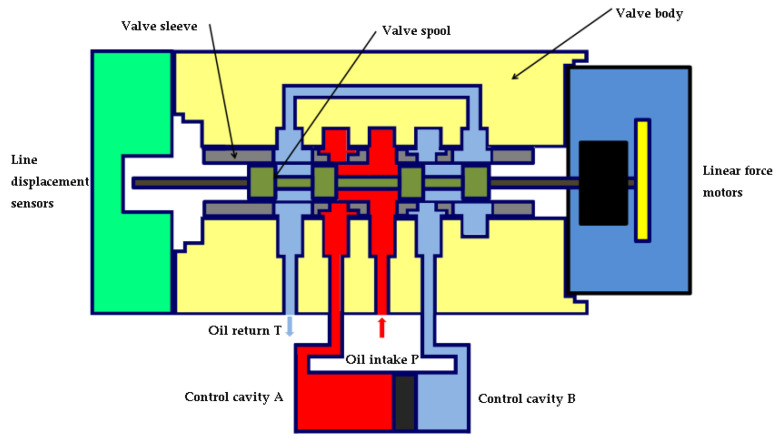
Spool moves to the left; left inlet port opens.

**Figure 4 sensors-22-07559-f004:**
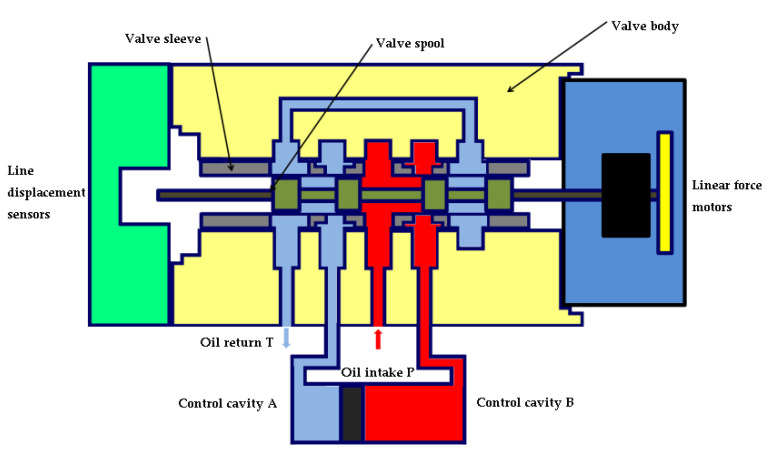
Spool moves to the right; right-hand inlet opens.

**Figure 5 sensors-22-07559-f005:**
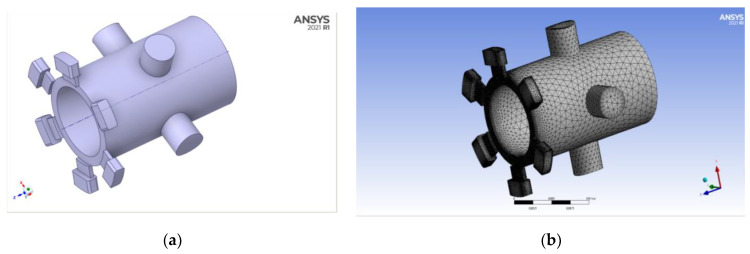
Simulated flow field model of spool valve sleeve: (**a**) Spool valve sleeve flow field model; (**b**) Flow field meshing model.

**Figure 6 sensors-22-07559-f006:**
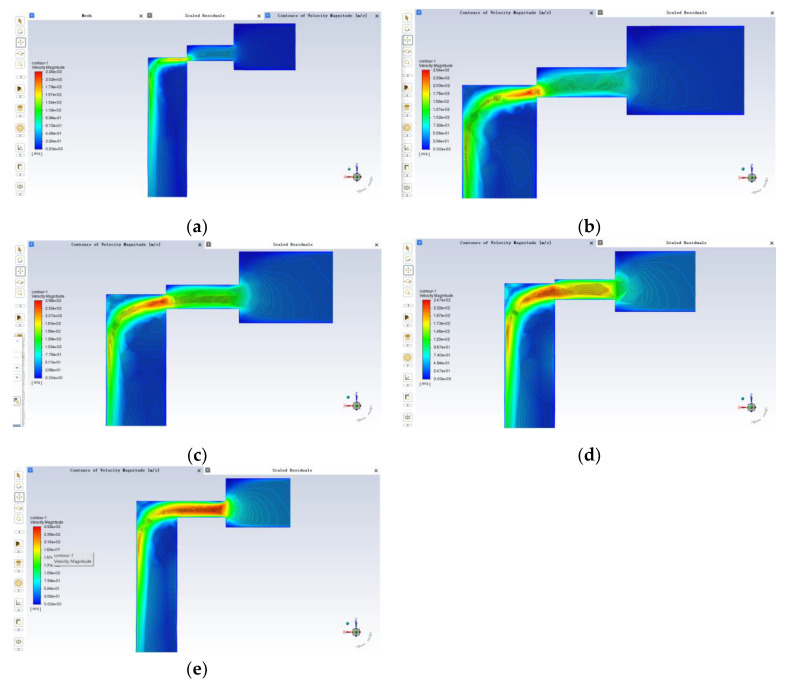
Valve port flow velocity analysis: (**a**) 0.1 mm; (**b**) 0.2 mm; (**c**) 0.3 mm; (**d**) 0.4 mm; (**e**) 0.5 mm.

**Figure 7 sensors-22-07559-f007:**
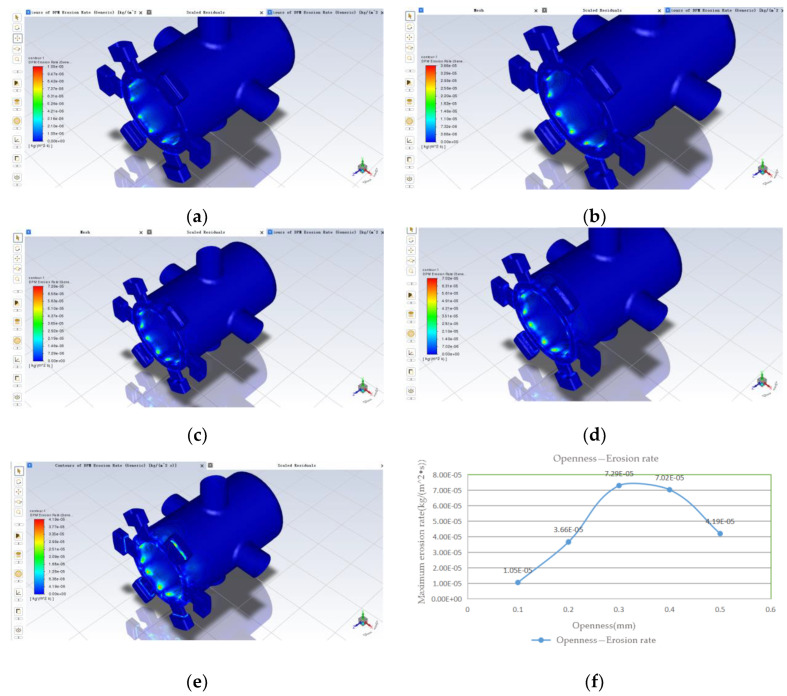
Effect of opening on erosion rate: (**a**) 0.1 mm; (**b**) 0.2 mm; (**c**) 0.3 mm; (**d**) 0.4 mm; (**e**) 0.5 mm; (**f**) variation of maximum erosion rate with increasing opening.

**Figure 8 sensors-22-07559-f008:**
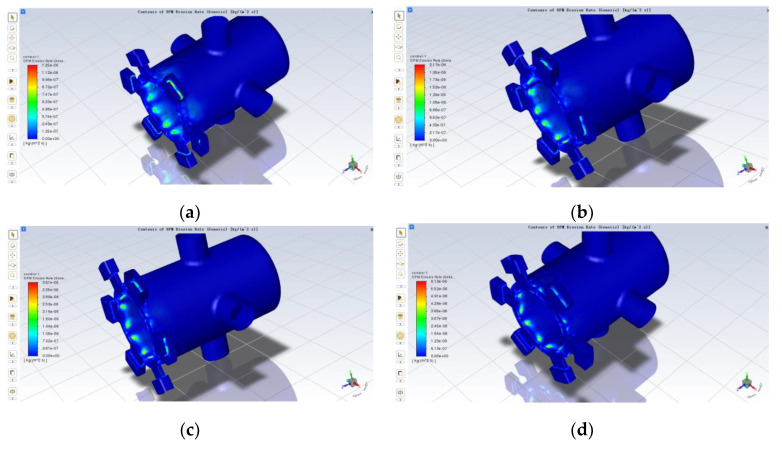
Effect of oil temperature on erosion rates: (**a**) 20 °C; (**b**) 40 °C; (**c**) 60 °C; (**d**) 80 °C; (**e**) 100 °C; (**f**) 120 °C; (**g**) graph of erosion rate as a function of temperature.

**Figure 9 sensors-22-07559-f009:**
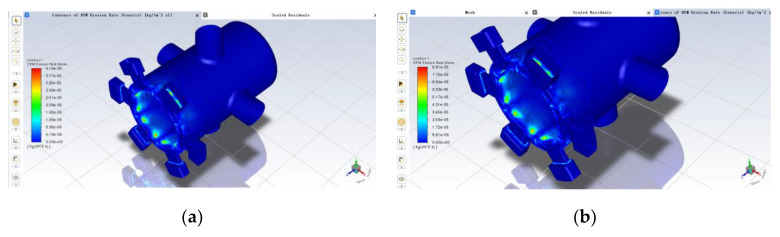
Effect of particle concentration on erosion rates: (**a**) original particle concentration; (**b**) 2 times particle concentration; (**c**) 4× particle concentration; (**d**) 8× particle concentration; (**e**) 16× particle concentration; (**f**) 32× particle concentration; (**g**) variation of maximum erosion rate with increasing particle concentration.

**Figure 10 sensors-22-07559-f010:**
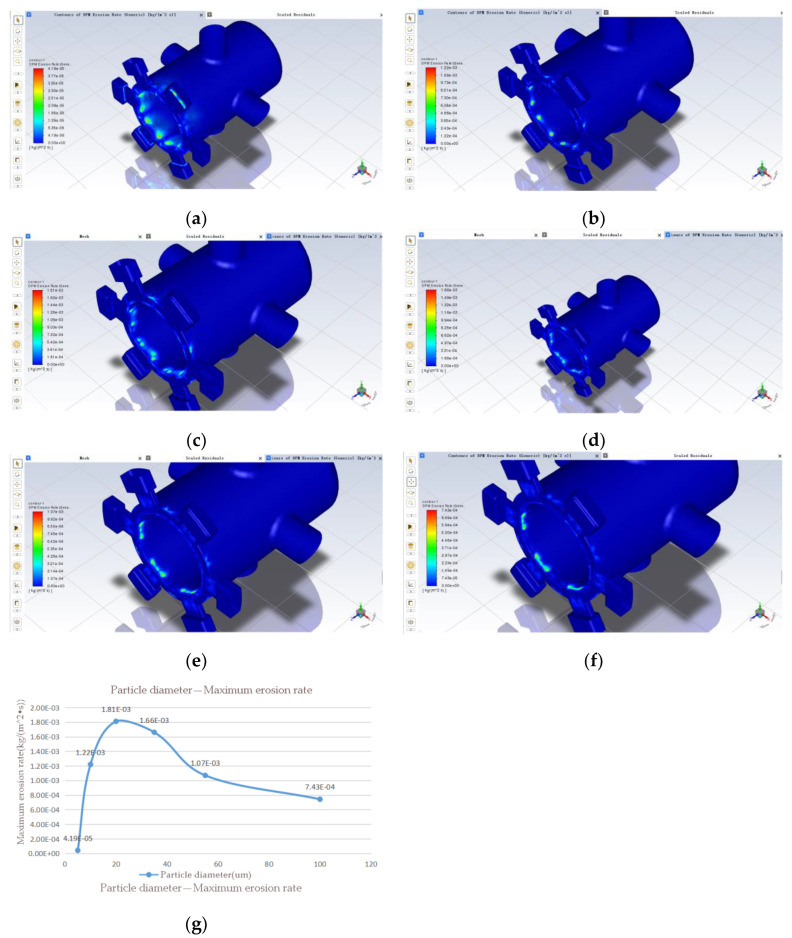
Effect of particle diameter on erosion rate: (**a**) 5 um; (**b**) 10 um; (**c**) 20 um; (**d**) 35 um; (**e**) 55 um; (**f**) 100 um; (**g**) variation of maximum erosion rate with increasing particle diameter.

**Figure 11 sensors-22-07559-f011:**
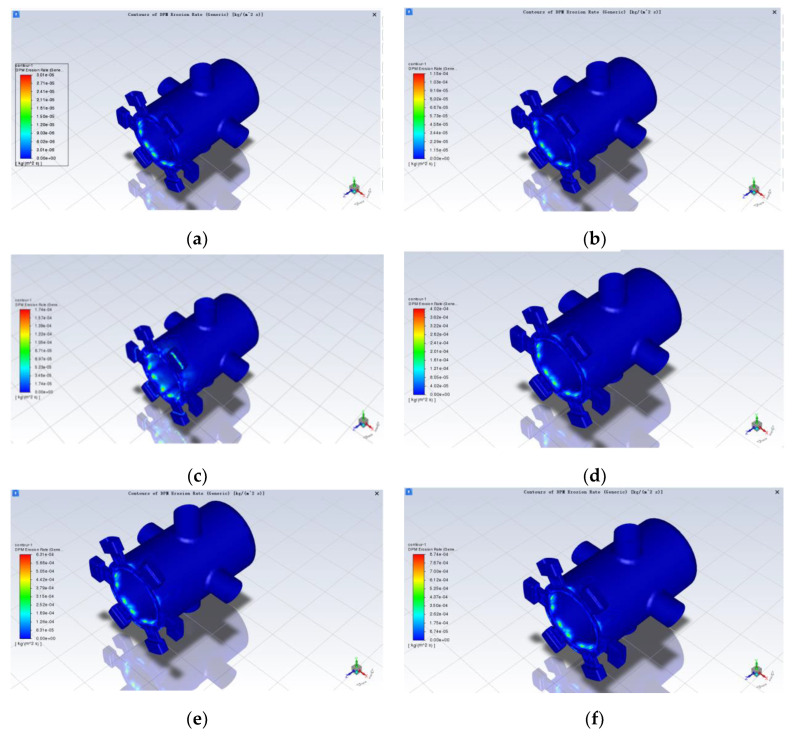
Effect of different pressure differentials on erosion wear: (**a**) 2 MPa; (**b**) 4 MPa; (**c**) 6 MPa; (**d**) 8 MPa; (**e**) 10 MPa; (**f**) 12 MPa; (**g**) 14 MPa; (**h**) curve of maximum erosion rate as a function of differential pressure.

**Table 1 sensors-22-07559-t001:** Impact angle function values.

Angle of impact *α/*°	0	20	30	45	90
Impact angle function *f*(*α*)	0	0.8	1	0.5	0.4

**Table 2 sensors-22-07559-t002:** Chinese YH-15 aviation hydraulic oil dynamic viscosity values at different temperatures.

Temperature/°C	Power Viscosity/Pa·s
−55	1.3859
−40	0.3342
−20	0.09496
0	0.03654
20	0.01922
40	0.01191
60	0.00828
80	0.00606
100	0.00477
120	0.00381
135	0.00344

## Data Availability

Data sharing is not applicable. No new data were created or analyzed in this study. Data sharing is not applicable to this article.

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
