# Peer review of "CFD-Based Physical Failure Modeling of Direct-Drive Electro-Hydraulic Servo Valve Spool and Sleeve"

_sensors, 2022, doi:10.3390/s22197559_

Round 1

Reviewer 1 Report

The manuscript entitled “CFD-based physical failure modeling of direct-drive electro-hy-2 draulic servo valve spool and sleeve” by Guoqin et al. discussed the internal motion characteristics of driver-driven servo valves and the erosion and provided the guidance for the structural optimization. Authors have established a mathematical model of erosion and wear of the spool sleeve for direct-drive analysis of servo valves. The erosion and wear effects of the servo valve spool sleeve were examined by various structural and flow parameters like opening degrees, contaminated particle diameters, and differential inlet and outlet pressures. All the figures are relevant and self explanatory. The experimental are performed with appropriate validatory approaches. I have few queries which need to be answered prior to acceptance of the manuscript:-

1.    Authors should incorporate an overall work flow chart of the study which will be highly beneficial for the future readers.

2.    There are several typos and mistakes. Author should thoroughly screen and revised the manuscript for typos and grammatical mistakes.

3.    Figures labellings in several figures are not clearly visible. So, author should share the high resolution images in the revised manuscript.

4.    Author should discuss the outcomes with earlier reported studies.

5.    Author should highlight the importance of the currently reported model as compared to the earlier reported models.

6.    Why authors have only considered for parameters to analyze the simulation?

7.    What are the other parameters which were not considered by authors?

Due to my above suggestions, I recommend minor revision of the manuscript.

Reviewer 2 Report

The content of the article titled: "CFD-based physical failure modeling of direct drive electrohydraulic servo valve spool and sleeve" has successfully studied the main failure modes of electrohydraulic servo valves directly, combined with erosion theory to calculate and discover the main failure modes of direct-drive electro-hydraulic servo valves.

The results got at the present time cannot be accepted for publication because there are no previous results to compare to confirm the accuracy of the results got, leading to the results got are quite ambiguous and very weak in terms of credibility. I can suggest that you do not accept the content of this article on the journal's system unless the authors must have almost completely rewritten it or have a way to other writing to enhance the evidence and attractiveness of the article content. In addition, English grammar and style need the author to review and supplement, as well as rewrite the entire introduction to clarify the title of the article.

Further comments:

In order to improve the current geometry, the author needs to restructure the entire section of the results and discussion by placing 3.1 characteristic factors, 3.2 influencing factors, in each part which are Theoretical results, part of simulation results need to be clearly delineated, and controlled so that readers can easily imagine, in addition, formulas for construction and citation from any source should be clearly stated.

In addition, it is necessary to rewrite the summary, introduction, and conclusion to clarify the content of the article, possibly changing the title to reveal the content of the article.

Reviewer 4 Report

This paper deals with very interesting topic - the numerical modelling of erosion wear inside the direct-drive electro-hydraulic servo valves under different flow conditions. These are some questions, remarks and recommendations:

1.         The paper requires English grammar corrections.

2.         Authors refer to 10 references. More than one half of these references are in Chinese and moreover difficult to download. Maybe some other relevant references could be offered to the readers.

3.         Line 125: This is an article, not a thesis.

4.         Lines 125-134: Authors refer to Forder formulas [8], but Equations (2)-(3) differ from the original work. Why? Authors should comment.

5.         Equation (4): Inconsistent index on the left-hand side.

6.         Line 147: Authors should state, what the function C(dp) is.

7.         Lines 150-151: “metal materials are plastic materials”… Should be specified more clearly.

8.         Figure 4, lines 257-259: Dimensions of the sleeve in Figure 4 are not visible. Basic dimensions of the sleeve should be given.

9.         Lines 239-254, 304-306, Table 2: More information about the computational mesh should be given. What is the size of the mesh? How does the cross-section look like? Does the computational domain represent full 3D geometry of the sleeve, or is there a symmetry plane? Also, boundary conditions should be described in more details. What are for example the thermal boundary conditions on the solid surfaces?

10.       Lines 248, 295-306: Authors consider changes of the dynamic viscosity with the changing temperature. What about changes of density, heat capacity and thermal conductivity? Authors should discuss.

11.       Sections 3.3.2, 3.5.1, 3.5.2, 3.6: Though these sections are ignoring the effect of temperature, the value of temperature must be given. 

Round 2

Reviewer 2 Report

The content of the article titled: "CFD-based physical failure modeling of direct-drive electrohydraulic servo valve spool and sleeve" has successfully studied the main failure modes of direct-drive, combined electro-hydraulic servo valves with erosion, theory to calculate and explore the main failure modes of direct-drive electro-hydraulic servo valves.

The results got at the present time have met the basic requirements to continue to edit and submit online. Authors need to complete the following requirements:

-Check all acronyms in the manuscript's content and fully add titles and labels for acronyms in all parts of the manuscript content.

-Adding more references from 2017 come back here, in order to increase the novelty of the manuscript's content and edit the references according to the journal's template.

-The author needs to clarify why, when choosing the sensitivity of hydraulic components, the author chooses the influencing factors as temperature and pressure. At each value of temperature and pressure, the author gives hydraulic characteristics at each specific location for easy reading. Can tabulate to follow. In addition, there should be serial sentences between sections as choosing which parameters are stable and which ones to change. When determining influencing factors, it is necessary to show how that factor directly affects it is necessary to have previously published results for verification. If not, the formula can confirm, after that, it is necessary to cite the specific formula. The formulas are cited from other sources, need to confirm.

-It is necessary to state the hydraulic nature of the servo valve. When the erosion increases or decreases, what causes it, should be added to the content to confirm the accuracy of the result.

Hopefully, the authors will fully supplement the requirements set out to accept the online submission.

Congratulations to the author on success with this useful work.

Author Response

Attached please find the file.

Reviewer 3 Report

Reviewer comments # 

I agree with the revisions for the manuscript. No more comments, the author has incorporated significant changes in the manuscript.
